# Down-Regulation of C1GALT1 Enhances the Progression of Cholangiocarcinoma through Activation of AKT/ERK Signaling Pathways

**DOI:** 10.3390/life12020174

**Published:** 2022-01-25

**Authors:** Juthamas Khiaowichit, Chutima Talabnin, Chavaboon Dechsukhum, Atit Silsirivanit, Krajang Talabnin

**Affiliations:** 1School of Translational Medicine, Institute of Medicine, Suranaree University of Technology, Nakhon Ratchasima 30000, Thailand; juthamas.khiaowichit@gmail.com; 2School of Chemistry, Institute of Science, Suranaree University of Technology, Nakhon Ratchasima 30000, Thailand; 3School of Pathology, Institute of Medicine, Suranaree University of Technology, Nakhon Ratchasima 30000, Thailand; chavaboon@sut.ac.th; 4Department of Biochemistry, Faculty of Medicine, Khon Kaen University, Khon Kaen 40002, Thailand; atitsil@kku.ac.th

**Keywords:** cholangiocarcinoma, core 1 β1-3 galactosyltransferase 1 (C1GALT1), cancer aggressiveness

## Abstract

Alteration of mucin-type O-glycosylation is implicated in tumor progression and metastasis of cholangiocarcinoma (CCA). Core 1 β1-3 Galactosyltransferase (C1GALT1) is a primary enzyme that regulates the elongation of core 1-derived mucin-type O-glycans. Dysregulation of C1GALT1 has been documented in multiple cancers and is associated with aberrant core 1 O-glycosylation and cancer aggressiveness; however, the expression of C1GALT1 and its role in CCA progression remains unknown. Our study demonstrated that C1GALT1 was downregulated in CCA tissues at both the mRNA and protein levels. The biological function of C1GALT1 using siRNA demonstrated that suppression of C1GALT1 in the CCA cell lines (KKU-055 and KKU-100) increased CCA progression, evidenced by: (i) Induction of CCA cell proliferation and 5-fluorouracil resistance in a dose-dependent manner; (ii) up-regulation of growth-related genes, ABC transporter genes, and anti-apoptotic proteins; and (iii) an increase in the activation/phosphorylation of AKT and ERK in silencing C1GALT1 cells. We demonstrated that silencing C1GALT1 in CCA cell lines was associated with immature core 1 O-glycosylation, demonstrated by high expression of VVL-binding glycans and down-regulation of other main O-linked glycosyltransferases β1,3-N-acetylglucosaminyltransferase 6 (*B3GNT6*) and ST6 N-Acetylgalactosaminide Alpha-2,6-Sialyltransferase 1 (*ST6GALNAC1*) in C1GALT1 knockdown. Our findings demonstrate that down-regulation of C1GALT1 in CCA increases the expression of immature core 1 O-glycan, enhancing CCA progression, including growth and 5-fluorouracil resistance via the activation of the AKT/ERK signaling pathway.

## 1. Introduction

Cholangiocarcinoma (CCA) is a lethal adenocarcinoma of the biliary epithelium [1]. The mortality for CCA varies worldwide, but a high incidence has been reported in East and Southeast Asia (China, Taiwan, South Korea, and Thailand) [2]. Liver fluke (*Opisthorchis viverrini*, OV) infection represents a substantial risk factor for CCA, while most CCAs have no identifiable cause [1]. CCA is characterized by an aberration of genetics, epigenetics, and post-translation modification [1]. Dysregulation of glycosylation has a significant role in the development and progression of CCA [3,4]. Aberrant glycosylations in CCA (e.g., mucin MUC5AC, MUC1, CCA-CA, carbohydrate antigen 19-9 (CA19-9), and CA-S27) have been used as proxies for CCA diagnosis as their altered expressions are correlated with the progression of CCA [4]. Mucins MUC1 and MUC5AC are strongly expressed in CCA and are related to the more aggressive phenotypes [5,6]. There is, however, no direct evidence regarding the underlying mechanisms responsible for the aberrant expression of the mucin-type O-glycan structure. The terminal GalNAc residue on the Tn antigen was identified using VVL—a lectin derived from the seeds of *Vicia villosa*—to stain binding glycans that are increased during cholangiocarcinogenesis [3]. In addition, mucin-type O glycans possess di- to hexa-saccharides with terminal galactose and sialic acids, which can be observed by direct structural analysis in CCA [7,8].

Core 1 β1, 3 galactosyltransferase I (C1GALT1) is a primary glycosyltransferase needed in the biosynthesis of mucin-type O-glycosylation. C1GALT1—with the aid of its C1GALT1-specific chaperone (COSMC)—transfers galactose to Tn-antigen (GalNAca-Ser/Thr) to form T-antigen (core 1 structure). T-antigen is further modified by adding other sugars via other glycosyltransferases to form the complex mucin-type O-glycans [9]. Dysregulation of C1GALT1 has been documented in various malignancies and has been linked to malignant phenotypes (i.e., cell proliferation and metastasis) and chemotherapeutic drug sensitivity [10,11,12,13]. The truncation of O-glycan has been associated with the genetic and epigenetic alterations of both C1GALT1 and COSMC, their specific chaperone [14,15,16]. Loss of C1GALT1 in *Kras* and *p53* mutant mice demonstrates the development of aggressive pancreatic ductal adenocarcinomas (PDACs) and increased metastasis. Additionally, knocking out C1GALT1 increases the truncation of O-glycosylation on MUC16, leading to increased tumorigenicity and PDAC aggressiveness [17]. A similar observation was reported in gastric cells, in which loss of C1GALT1 activity is the cause of gastritis and gastric cancer by caspase-1/caspase-11 (Casp1/11)–dependent inflammasomes [18]. Sagar et al. demonstrated that disruption of COSMC contributes to an increase in truncated O-glycans on MUC4, which enhances gemcitabine resistance in PDAC tumors by altering ErbB/AKT signaling [19]. C1GALT1 over-expression—associated with tumor growth, metastasis, and poor prognosis—has been observed in various cancers—viz., ovarian cancer [20], head and neck cancer [13], hepatocellular carcinoma [21], breast cancer [10], colorectal cancer [22], and gastric cancer [12]. Silencing C1GALT1 expression inhibits cancer progression by blocking O-glycan elongation on several growth receptors—e.g., EGFR [13], MET receptor [21], FGFR2 [22], β1-integrin [12,23,24]. Taken together, dysregulation of C1GALT1 is involved in cancer development and progression by promoting either O-glycan truncation or elongation. This evidence motivated us to explore the significance of C1GALT1 in CCA. We sought to determine C1GALT1 expression in CCA tissues and CCA cell lines. The biological effects of C1GALT1 on CCA cell growth, 5-fluorouracil sensitivity, and its underlying mechanism were studied.

## 2. Materials and Methods

### 2.1. Chemicals and Antibodies

Cell culture reagents, including Dulbecco’s Modified Eagle Medium (DMEM; cat. no. 12100-046), fetal bovine serum (FBS; cat. no. 10270-098), and penicillin/streptomycin (cat. no. 15140-122), were obtained from Thermo Fisher Scientific, Inc. Specific siRNA to C1GALT1 (siC1GALT1; cat. no. sc-72690) and non-targeted negative control siRNA (siControl; cat. no. sc-37007) were purchased from Santa Cruz Biotechnology, Inc. DharmaFECT^®^ transfection reagent (cat. no. T-2001-20) was obtained from GE Healthcare Dharmacon, Inc. Cell counting kit 8 (CCK-8; cat. no. ab228554) was purchased from Abcam, Inc. TRIzol reagent (cat. no. 15596026) was obtained from Thermo Fisher Scientific, Inc. SensiFAST cDNA Synthesis Kit (cat. no. BIO-65053) was purchased from Bioline; Meridian. LightCycler^®^ 480 SYBR green I master mix in SYBR-Green (cat. no. 04707516001 was purchased from Roche Molecular Systems, Inc. Liquid DAB+ (cat. no. #8059) was purchased from Cell Signaling Technology, Inc. Immobilon Forte Western HRP substrate (cat. no. WBLUF0100) was obtained from Merck Millipore; Darmstadt; Germany), and ABC-Peroxidase Solution (cat. no. PK-4000) was purchased from Vector Laboratories, Inc. Primary antibodies against C1GALT1 (42 kDa) (cat. no. sc-100745; 1:500-1000) and β-actin (42 kDa) (cat. no. sc-47778; 1:2000) were purchased from Santa Cruz Biotechnology, Inc. Antibodies against total AKT (60 kDa) (cat. no. 4685S; 1:1000), ser 473 phosphorylated AKT (pAKT; 60 kDa) (cat. no. 4060S; 1:1000), total ERK (42 kDa) (cat. no. 4685S; 1:1000), Thr202/Tyr204 phosphorylated ERK (pERK; 42, 44 kDa) (cat. no. 4060S; 1:1000), BAX (21 kDa) (cat.no. 50599-2; 1:1000), BCL2 (~27 kDa) (cat.no. 12789-1; 1:1000) were purchased from Cell Signaling Technology, Inc. Biotinylated lectin kit I (cat. no. BK-1000), including biotinylated peanut agglutinin (PNA; 1:100), wheat germ agglutinin (WGA; 1:1000), vicia villosa lectin (VVL; 1:1000) were obtained from Vector Laboratories, Inc.

### 2.2. Gene Expression Analysis from the Database

Gene expression data (GEO series GSE76297) was obtained from the Gene Expression Omnibus (GEO) database (https://www.ncbi.nlm.nih.gov accessed on 5 June 2020). GEO series GSE76297 comprises expression data from 90 CCA tumors and matched paired non-tumors. Tumors and paired non-tumor tissues were analyzed individually utilizing the Affymetrix Human Transcriptome Array 2.0 for gene expression profiling. All expression data were log2 transformed. Fold differences were calculated and presented using the ratio of each gene between tumors and paired non-tumors. Fold changes were categorized: <1.5-fold = “low expression” and >1.5-fold = “high expression”.

### 2.3. Human CCA Tissues and Cell Lines

The paraffin-embedded tissues from 26 CCA patients and 29 frozen CCA tissues and matched adjacent tissues were obtained from the Liver Fluke and Cholangiocarcinoma Research Center of Khon Kaen University Thailand. Informed written consent was obtained from all patients, and the Ethics Committee for Human Research from Khon Kaen University and Suranaree University of Technology reviewed and approved the procedure for sample collection (HE521209 and EC-57-25, respectively).

Four CCA cell lines—KKU-055, KKU-100, KKU-213A, and KKU-213B—were established [25]. In addition, immortalized cholangiocytes (MMNK-1) were provided by Dr. Sopit Wongkham. Certificates of authentication were obtained from the Japanese Collection of Research Bioresources Cell Bank in Osaka, Japan. All cell lines were cultured in Dulbecco’s Modified Eagle Medium (DMEM) supplemented with 10% fetal bovine serum (FBS) and 1% penicillin/streptomycin. Cells were incubated at 37 °C with 5% CO_2_. Mycoplasma testing was conducted for all cell lines using Hoechst 33258 DNA staining (cat.no. H3569, Thermo Fischer Scientific, Waltham, MA, USA) and PCR detection.

### 2.4. RNA Extraction and Quantitative Real-Time PCR (qPCR)

According to the manufacturer’s protocol, total RNA from frozen human CCA tissues and cell lines was extracted by TRIzol reagent and subsequent cDNA synthesis using the SensiFAST cDNA Synthesis Kit. Quantitative PCR reactions were performed with a LightCycler^®^ 480 SYBR green I master mixer. Primer sets are listed in Table 1. The gene amplification condition was performed as previously described with a slight modification to the annealing temperature [26]. The annealing temperature was: 55 °C for *ST6GALNAC1*, *COSMC*, *CDK4*, *CCNE1*, and *CCNB1*; 58 °C for *C1GALT1*, *B3GNT6*, *C*-*Myc*, *BIRC5*, *ABCC1*, *ABCC3*, and *ABCG2*; and 60 °C for *CCND1*. *β*-*Actin* was used as the internal control to normalize the expression of the target genes. Relative mRNA expression was calculated using the 2^−^^ΔΔCt^ method [27], and 1.5-fold changes were used as the cut-off value. In the statistical analysis, <1.5-fold changes were categorized as “low expression”; and >1.5-fold changes as “high expression”.

### 2.5. Immunohistochemistry

Formalin-fixed paraffin-embedded (FFPE) tissues of CCA were cut into 5-µm-thick sections. Immunohistochemical staining was performed according to previous confirmed experimental procedures [28]. In brief, each section was first deparaffinized and rehydrated, then boiled for 5 min (in 0.01 mol/L citrate buffer at pH 6.0) in a pressure cooker for epitope retrieval. The endogenous peroxidase was inactivated by 3% hydrogen peroxide in methanol for 30 min. Normal horse serum (dilution 1:20) was used for another 30 min to block non-specific binding. The sections were incubated with primary mouse anti-human C1GALT1 monoclonal antibody (dilution 1:100) at room temperature overnight. After washing with PBS, each section was incubated with a biotinylated secondary antibody (1 µg/mL) in PBS for 1 h at room temperature, then incubated with ABC-Peroxidase Solution for 1 h at room temperature. The visualization using 3,3′-diaminobenzidine-tetrahydrochloride, Liquid DAB+, and counterstaining with hematoxylin were performed as per the manufacturer’s protocol. 

The intensity of C1GALT1 expression in tumor cells was scored as follows: 0 = no staining; 1+ = light brown; and 2+ = intense brown. Without knowing the prognosis or clinicopathological variables, two researchers evaluated the samples using an Olympus microscope with ×100 magnification. A kappa value of 0.752 indicated a good agreement of the data presented by the two different sources, thereby validating our evaluation method. A score of 0 was categorized as C1GALT1-negative expression, while a score of 1 or 2 was categorized as C1GALT1-positive expression. 

### 2.6. C1GALT1 Knockdown 

C1GALT1-knockdown was performed using small interfering RNA (siRNA) in the CCA cell lines, KKU-055, and KKU-100. The cells—at a density of 2.5 × 10^5^ cells/well—were seeded into a 6-well plate and incubated overnight. According to the manufacturer’s protocol, cells were transfected with 10 µM siControl or siC1GALT1 using DharmaFECT^®^ transfection reagent. After transfection for 48 h, transfected cells were used for determining C1GALT1 mRNA and protein expression and subsequent experiments.

### 2.7. Cell Viability 

Transfected cells (1 × 10^3^ cells/well) were seeded into 96-well plates then incubated at 37 °C for 0, 1, 2, 3, 4, and 5 days. The culture medium was changed every second day. According to the manufacturer’s instructions, cell viability was measured using a cell counting kit 8 (CCK-8). In brief, 10 µL of CCK-8 solution was added to each well and incubated at 37 °C for 1 h. Color development in this assay was measured at 460 nm using a microplate spectrophotometer (Bio-Rad Laboratories, Inc., Hercules, CA, USA). In addition, the proliferative rate (Relative to day 0) was calculated.

### 2.8. 5-Fluorouracil (5-FU) Treatment

Transfected cells (5 × 10^3^ cells, 100 µL/well) were seeded into 96-well plates. After 24 h seeding, the 5-FU concentrations with 30–50% cytotoxicity (150 and 300 µM) were used to treat the cells for 48 h. Cell viability was detected using the CCK-8 kit. The relative cell number (Relative to 0 µM) was calculated.

### 2.9. Protein Collection and Western Blot Analysis

After 48 h of transfection, transfected cells were washed and subsequently treated with lysis buffer [26]. The protein concentration of the whole-cell lysate was measured using a Pierce^®^ BCA protein assay kit (cat no. 23225; Thermo Fisher Scientific, Inc.). Protein samples (30 µg/lane) were then separated using 10% SDS-polyacrylamide gel electrophoresis then transferred to nitrocellulose membranes using wet/tank electroblotting techniques. Immunodetection of target proteins was achieved using primary antibodies: Anti-β-actin, anti-C1GALT1, anti-total AKT, anti-pAKT (Ser473), anti-total ERK, anti-pERK (Thr202/Tyr204), anti-BAX, and anti-BCL2 at 4 °C overnight. Immunoreactivity of the target proteins was detected using an Immobilon Forte Western HRP substrate. The signal intensities were quantified using ImageJ software (version 1.53a; National Institutes of Health). The relative protein expression was normalized against β-actin. The respective ratios of each protein to β-actin were calculated, including the ratios of pAKT/AKT, pERK/ERK, and BCL2/BAX.

### 2.10. Lectin-Cytochemistry

After 48 h of transfection, transfected cells (6 × 10^4^ cells/well) were seeded into the 24-well plate overnight. Cells were then fixed with 4% paraformaldehyde in PBS (pH 7.4 for 15 min) and permeabilized with 0.2% Triton X-100 in PBST for 10 min. Next, non-specific binding was blocked with 0.3% of FBS Fetal Bovine Serum) for 30 min. After blocking, cells were incubated with biotinylated peanut agglutinin (PNA,1:100), wheat germ agglutinin (WGA, 1:1000), and *Vicia villosa* lectin (VVL, 1:1000) at 4 °C overnight. The ABC-Peroxidase Solution was used to develop the signal for 1 h at room temperature. The visualization with 3, 3′-diaminobenzidine-tetrahydrochloride, Liquid DAB+, and counterstained with hematoxylin were then performed. The images were captured by the Zeiss microscope system with original magnification, ×200.

### 2.11. Statistical Analysis

SPSS statistics software version 16.0.1 was used for all statistical analyses (SPSS, Chicago, IL, USA). The over-survival curve was estimated using the Kaplan–Meier survival analysis, and the log-rank test was performed to compare groups. The results from functional investigations of C1GALT1—including cell viability, 5-fluorouracil treatment, and mRNA and protein expression—were performed in three independent experiments then were expressed as the mean ± SEM. The Student’s *t*-test was used to test the difference between the two groups. A *p*-value < 0.05 was accepted as statistically significant.

## 3. Results

### 3.1. Expression of O-Linked Glycosyltransferases in CCA Tissues

The initiation of mucin-type O-glycosylation occurs via the formation of Tn antigen, then the modification process of mucin-type O-glycans is directly regulated by different glycosyltransferases [9]. Therefore, in order to investigate whether aberrant mucin-type O-glycosylation occurs in CCA, the differential expression of O-linked glycosyltransferases (including *C1GALT1*, *COSMC*, *B3GNT6*, and *ST6GALNAC1*) was investigated through GEO series GSE76297. Using a cut-off of 1.5-fold change, the expression levels of these four genes were down-regulated in CCA tissues compared to non-tumor tissues, evidenced by the percent of tumor cases showing high expression of *C1GALT1*–29% (26/90), *COSMC*–7% (6/90), *B3GNT6*–7% (15/90), and *ST6GALNAC1*–24% (22/90) (Figure 1A–D). These findings suggest that altered expression of O-linked glycosyltransferases occurs in CCA.

### 3.2. Down-Regulation of C1GALT1 in CCA Tissues

The synthesis of core 1-derived mucin-type O-glycans is a crucial precursor for the complex formation of mucin-type O-glycans [9]. We, therefore, verified the differential expression of C1GALT1 at both the mRNA and protein levels using qPCR and immunohistochemistry. The mRNA expression experiments demonstrated that there were 23% (7/30) of tumor cases showing a high expression of *C1GALT1*, whereas 77% (23/30) showed a low expression compared with the adjacent normal tissue using a cut-off value of 1.5-fold change (Figure 2A). This finding is consistent with the study of C1GALT1 protein expression in CCA tissues by immunohistochemistry, in which 38% (10/26) had positive expression while the remaining 62% (16/26) had negative expression (Figure 2B).

Subsequently, CCA patients were dichotomized into two groups based on protein expression results, C1GALT1-negative and -positive expression. There was no statistically significant association between C1GALT1 protein expression with any clinicopathological features of CCA patients (data not shown). The Kaplan–Meier analysis and log-rank test revealed no statistically significant difference in overall survival between patients with C1GALT1-positive expression and C1GALT1-negative expression (log-rank, *p*-value = 0.342) (Figure 2C). However, the mean survival time of the CCA patients with C1GALT1-negative expression (19 weeks) was less than that of the patients with C1GALT1-positive expression (37 weeks). These findings suggest that down-regulation of C1GALT1 may be associated with a poor prognosis of CCA patients.

### 3.3. Silencing C1GALT1 Promotes CCA Cell Proliferation and 5-Fluorouracil (5-FU) Drug Resistance

In order to address the biological functions of C1GALT1 in CCA, endogenous C1GALT1 expression was examined in four CCA lines (KKU-055, KKU-100, KKU-213A, and KKU-213B) and an immortalized cholangiocyte (MMNK-1) using western blot analysis. High expression of C1GALT1 was demonstrated in KKU-055, KKU-100, and MMNK-1, whereas KKU-213A and KKU-213B—highly aggressive phenotypes—had a relatively low endogenous expression of C1GALT1 (Figure 3A). As a result, KKU-055 and KKU-100 cells were used in a knockdown model because of their higher endogenous C1GALT1 expression than KKU-213A and KKU-213B. KKU-055 and KKU-100 cells were transfected with specific siRNA to C1GALT1 (siC1GALT1) and siControl. Furthermore, the mRNA and protein expression levels of C1GALT1 were significantly decreased following transfection with C1GALT1 siRNA at 48 h (Figure 3B,C). In order to understand the role of C1GALT1 in CCA, we further investigated whether suppression of C1GALT1 influences CCA cell growth and 5-FU sensitivity. The cell proliferation assay demonstrated that silencing C1GALT1 increased cell growth from day 2 to day 5 in both KKU-055 and KKU-100 cells (Figure 3D) and promoted 5-FU resistance at 150 and 300 µM, in KKU-055 and KKU-100, respectively (Figure 3E). Moreover, the effect of silencing C1GALT1 on CCA progression was demonstrated by up-regulation of growth-related genes (*CCND1*, *CDK4*, *CCNE1*, and *c-Myc*) and the ABC transporter superfamily (*ABCC1*, ABCC3 and ABCG2) (Figure 3F). These findings suggest that C1GALT1 has a role in the regulation of CCA cell growth and 5-FU response.

### 3.4. Silencing C1GALT1 Increases CCA Progression via the Activation of the AKT/ERK Signaling Pathway

In order to evaluate the underlying mechanism involved in silencing C1GALT1-mediated CCA progression, C1GALT1 expression was suppressed using siC1GALT1 or siControl in KKU-055 and KKU-100 cells, and survival markers including BCL2, AKT, and ERK were determined. First, western blot analysis confirmed up-regulation of anti-apoptotic protein (BCL2) and down-regulation of apoptotic protein (BAX) after C1GALT1 suppression (Figure 4A). Then, the effect of silencing C1GALT1-mediated CCA survival was demonstrated by a high ratio of BCL2/BAX (Figure 4A). The activation/phosphorylation of AKT and ERK was significantly increased in silenced C1GALT1 cells (Figure 4B,C). These findings indicate that inhibition of C1GALT1 induces AKT/ERK activation, leading to CCA cell growth and survival.

### 3.5. Silencing C1GALT1 Is Associated with the Truncation of Core 1-Derived Mucin-Type O-Glycans

The relative expression of the C1GALT1, B3GNT6, and ST6GALNAC1 enzymes towards the Tn-antigen (GalNAca-Ser/Thr) determines the variable synthesis of mucin-type O-glycan extension or truncation [9]. In order to understand the effect of C1GALT1 knockdown on mucin-type O-glycosylation in CCA, the respective mRNA expression of *C1GALT1*, *COSMC*, *B3GNT6*, and *ST6GALNAC1* was determined after transfection with C1GALT1 siRNA at 48 h. A decrease in *ST6GALNAC1* and *B3GNT6* was detected after silencing siC1GALT1 cells compared to control cells. Whereas no transcription levels changed vis-à-vis the private C1GALT1 chaperone, *COSMC* was detected after C1GALT1 suppression (Figure 5A). Upon C1GALT1 suppression, the aberrant mucin-type O-glycosylation was further investigated using lectin-cytochemistry. The respective binding specificity of GalNAc-α Thr/Ser, Gal-β-(1,3)-GalNAc-α-Thr/Ser and GlcNAc; Neu5Ac was determined using the *Vicia villosa lectin* (VVL), peanut agglutinin (PNA), and wheat germ agglutinin (WGA). The results revealed high VVL-binding glycans expression and low WGA-binding glycans expression in silenced C1GALT1 cells when compared to control cells. There was, however, no difference in the expression of PNA-binding glycans between the silenced C1GALT1 and control cells (Figure 5B). The inhibition of C1GALT1, therefore, diminished *B3GNT6* and *ST6GALNAC1* expression, leading to mucin-type O-glycans truncation in CCA.

## 4. Discussion

Alteration of mucin-type O-glycosylation is implicated in every critical step of tumor formation and progression [29]. The most frequently observed aberrant mucin-type O-glycosylation in cancer is the immature truncated core 1 O-glycans, which are designated as the Tn and sialyl -Tn (sTn) antigens [30,31,32,33]. Tn-antigen is a common feature of CCA, in which enhanced expression is shown during cholangiocarcinogenesis and is associated with the expression of polypeptide N-acetylgalactosaminyl-transferases, isoform 5 (*GALNT5*) [19]. The underlying mechanism of Tn antigen expression and its modification process remain unclear. Altered expression of O-linked glycosyltransferases is crucial in determining the variable modification of glycosylation on the Tn antigen. We showed that low mRNA expression of O-linked glycosyltransferases (*C1GALT1*, *B3GNT6*, and *ST6GALNAC1*) was detected in CCA tissues, suggesting that the high expression of the Tn antigen in CCA may be due to the lack of activity of these three glycosyltransferases vis-à-vis promoting Tn antigen modification.

C1GALT1 is a key glycosyltransferase that promotes the complexity of glycosylation on the Tn antigen by producing Core-1 glycans, which act as precursors for forming the complex mucin-type O-glycans [9]. Down-regulation of C1GALT1 was verified in CCA tissues at both the mRNA and protein levels. These results suggest a crucial role for C1GALT1 in the aberrant expression of the Tn antigen in CCA. The findings are consistent with studies of various cancers, in which the loss of C1GALT1 and their specific chaperones, COSMC, contribute to increased truncation of O-glycans on several mucin proteins, including MUC4 and MUC16 [17,19]. The truncation of the mucin-type O-glycan has been associated with the genetic and epigenetic alterations of C1GALT1 and their specific chaperone, COSMC [14,15,16,34]. As a result, further studies on the genetic and epigenetic alterations of C1GALT1 and COSMC in CCA are required to ascertain the underlying mechanism of C1GALT1 down-regulation. In contrast, the up-regulation of C1GALT1 is also observed in various cancers and is associated with poor survival and tumor progression. These conflicting data among the different types of cancer suggest a specific role for C1GALT1 in specific cellular contexts such as growth activation, migration, and invasion capabilities.

Loss of C1GALT1 function is a cause of the development and progression of various cancers. We demonstrated that suppression of C1GALT1 in CCA cell lines contributed to various CCA aggressive malignant phenotypes, including cell growth and 5-FU resistance through increased transcription levels of growth-related genes (*CCND1*, *CDK4*, *CCNE1*, and *c**-MYC*) and the ABC transporter superfamily (*ABCC1*, *ABCC3*, and *ABCG2*). Subsequently, inhibition of C1GALT1 enhanced CCA progression via up-regulation of anti-apoptotic protein (BCL2) and activation/phosphorylation of AKT and ERK. These findings agree with studies on colorectal cancer, gastric cancer, and pancreatic ductal adenocarcinoma, in which the knockout of C1GALT1 in cancer cells significantly induced Tn-antigen expression and subsequently enhanced cell proliferation, adhesion, as well as migration and invasiveness [11,17,18]. Moreover, the knockout of the C1GALT1-specific chaperone (COSMC) increases truncated O-glycan on MUC4, enhancing malignant phenotypes and gemcitabine resistance in PDAC tumors by altering the ErbB/AKT signaling cascades [19]. Taken together, the current observations highlight the significant role of C1GALT1 down-regulation in CCA cell growth and 5-FU resistance via the AKT/ERK signaling pathway. However, the effect of C1GALT1 in CCA progression requires further study vis à vis CCA metastasis, its sensitivity to other chemotherapeutic drugs and resistance mechanism that trigger the significant role of C1GALT1 down-regulation in CCA tumor progression.

The formation of Tn-antigen is mediated via the action of polypeptide N-Acetylgalactosaminyl-transferase (GALNTs). A previous study demonstrated that the expression level of GALNT5 in CCA corresponds to the expression level of Tn-antigens as detected by Tn-specific *Vicia villosa* lectin (VVL-binding glycans). Additionally, over-expression of GALNT5 in CCA cell lines promotes the expression level of VVL-binding glycans, resulting in cell invasion and metastasis through the activation of the AKT/ERK signaling pathway in CCA [3]. However, the Tn antigen can be further modified by the action of C1GALT1 to promote the formation of the core 1 O-glycans (T-antigen). The suppression of C1GALT1 increases the expression level of VVL binding glycans in CCA cell lines. These suggest that the high expression of Tn-antigen in CCA could result from the down-regulation of C1GALT1. The relative expression of C1GALT1, B3GNT6, and ST6GALNAC1 enzymes determines the variable modification of glycosylation on the Tn antigen. The competition between these three enzymes was tested by selective suppression of C1GALT1, resulting in increased expressions of sialyl-Tn and GSL-II binding (Core-3 glycans) in human colon cancer cells [35]. In contrast, the reduction of *B3GNT6* and *ST6GALNAC1* mRNA expression was detected in CCA cell lines with C1GALT1 knockdown compared to control cells. This result suggests that the lack of C1GALT1 in CCA may sustain the expression level of the Tn-antigen as detected by the Tn-specific *Vicia villosa* lectin.

The present study has limitations. The contrasting result between the C1GALT1 over-expression model and the in vivo experiment means that the role of C1GALT1 on CCA progression needs better definition. In addition, further studies are needed on (i) C1GALT1 expression in large sample size and clinical significance including progression-free survival and disease-free survival, and (ii) the target glycoprotein of C1GALT1 in CCA.

In conclusion, our study demonstrated the down-regulation of C1GALT1 in CCA. Silencing C1GALT1 enhanced CCA cell growth and 5-FU resistance by activating the AKT/ERK signaling pathway. Suppression of C1GALT1 seems to sustain the expression level of VVL-binding glycans in the CCA cell line by reducing the expression of Core-3 synthase (*B3GNT6*) and sialyl-transferase (*ST6GALNAC1*) expression. The loss of C1GALT1 function thus contributed to the formation of the immature truncated core 1 O-glycan (Tn-antigen), leading to the CCA progression and chemoresistance through the activation of the AKT/ERK signaling pathway.

## Figures and Tables

**Figure 1 life-12-00174-f001:**
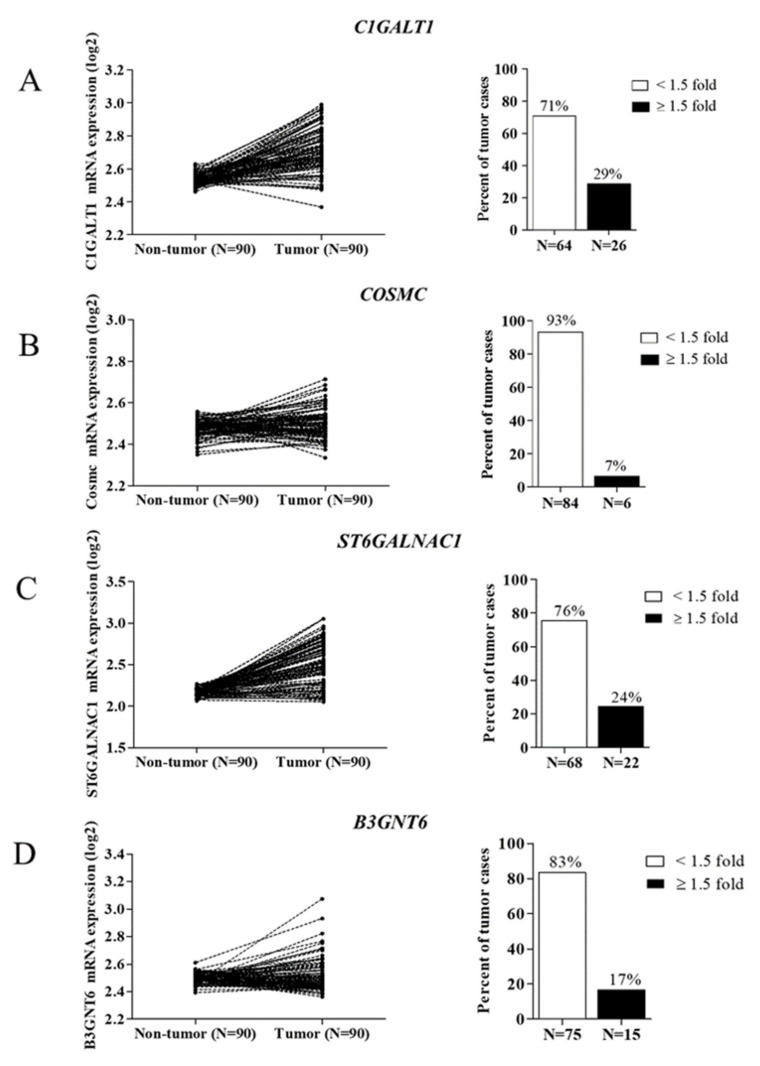
Expression levels of O-linked glycosyltransferases in CCA. The mRNA expression levels of (**A**) *C1GALT1*, (**B**) *COSMC*, (**C**) *ST6GALNAC1,* and (**D**) *B3GNT6* in 90 paired CCA tissues were obtained from the GEO database (GEO series GSE76297).

**Figure 2 life-12-00174-f002:**
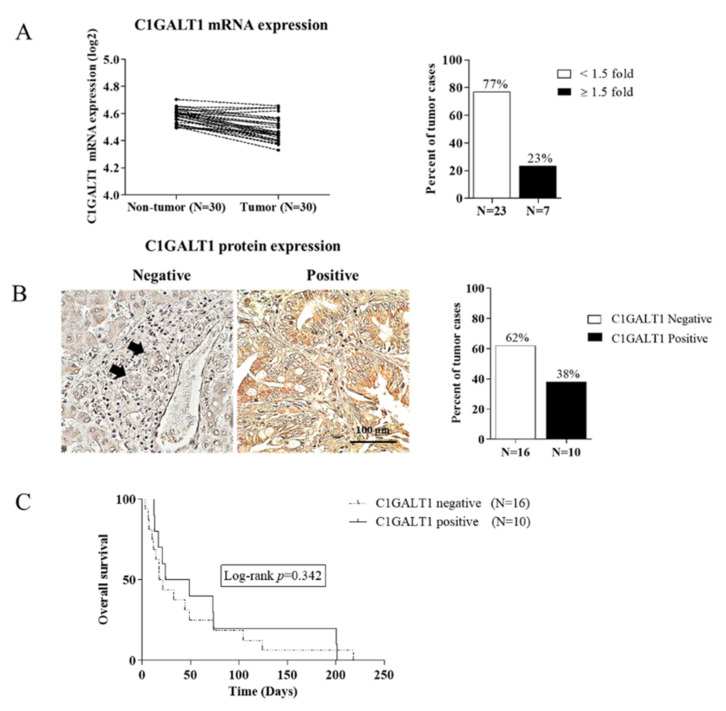
C1GALT1 is down-regulated in CCA tissues. (**A**) The *C1GALT1* mRNA expression was determined in 29 paired frozen CCA tissues by qPCR. Relative mRNA expression level was normalized to β-actin as the reference genes and calculated using the 2^−ΔΔCt^ method. Values are expressed as the mean ± SEM of three independent experiments. (**B**) Immunohistological analysis of C1GALT1 protein expression in 26 CCA tissues (Left panel, ×400 magnification); Scale bar, 100 µm and percentage (%) of tumor cases for C1GALT1 expression in CCA tissues (Right panel). (**C**) Kaplan–Meier analysis demonstrated cumulative overall survival determined of CCA patients with C1GALT1 negative VS C1GALT1 positive.

**Figure 3 life-12-00174-f003:**
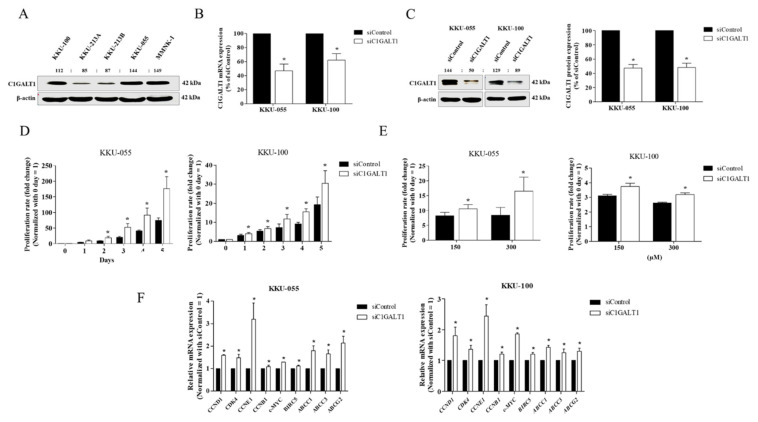
Effect of silencing C1GALT1 on malignant phenotypes. (**A**) Basal protein expression of C1GALT1 in 4 CCA cell lines (KKU-100, KKU-213A, KKU-213B and KKU-055) and an immortalized human cholangiocyte cell line (MMNK-1). KKU-055 and KK-100 cells were treated with siC1GALT1 and siControl for 48 h. (**B**,**C**) The efficiency of C1GALT1 knockdown was determined using qPCR and western blot analysis. (**D**) The effect of C1GALT1 suppression on CCA cell proliferation was determined at day 1 to day 5 using the CCK-8 kit. (**E**) The effect of C1GALT1 suppression on the 5-FU response, KKU-055 and KKU-100 cells were treated with 5-FU at 150 and 300 µM for another 48 h. Proliferation rates were presented as mean ± SEM of three independent experiments. * *p* < 0.05 VS day 0 or * *p* < 0.05 VS siControl. (**F**) The relative mRNA expression of growth-related genes, ABC transporter genes and anti-apoptotic related gene were determined using qPCR and calculated using the 2^−ΔΔCt^ method. Expression values were presented as mean ± SEM of three independent experiments. * *p* < 0.05 VS siControl.

**Figure 4 life-12-00174-f004:**
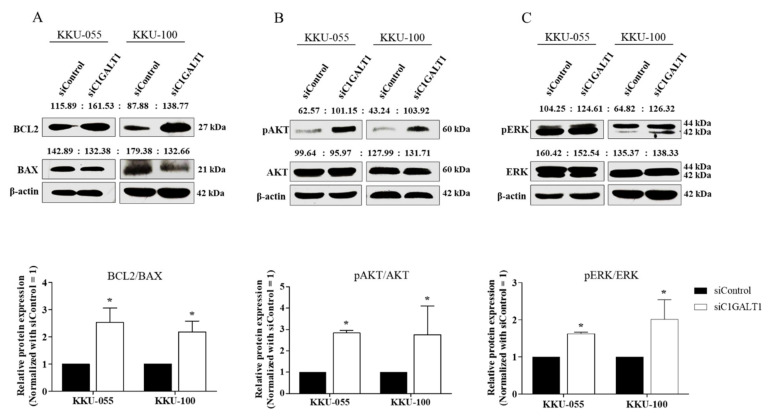
Silencing C1GALT1 increases the activation of the AKT/ERK signaling pathway. KKU-055 and KKU-100 cells were treated with siC1GALT1 and siControl for 48 h. The protein expression levels of (**A**) anti-apoptotic proteins (BCL2, and BAX), (**B**) AKT, and (**C**) ERK were determined using western blot analysis and compared between the C1GALT1 knockdown and control cells. β-actin was used as the internal control. Relative protein expression was measured by Image J software (version 1.53a). Expression values are calculated as mean ± SEM of three independent experiments. * *p* < 0.05 VS siControl.

**Figure 5 life-12-00174-f005:**
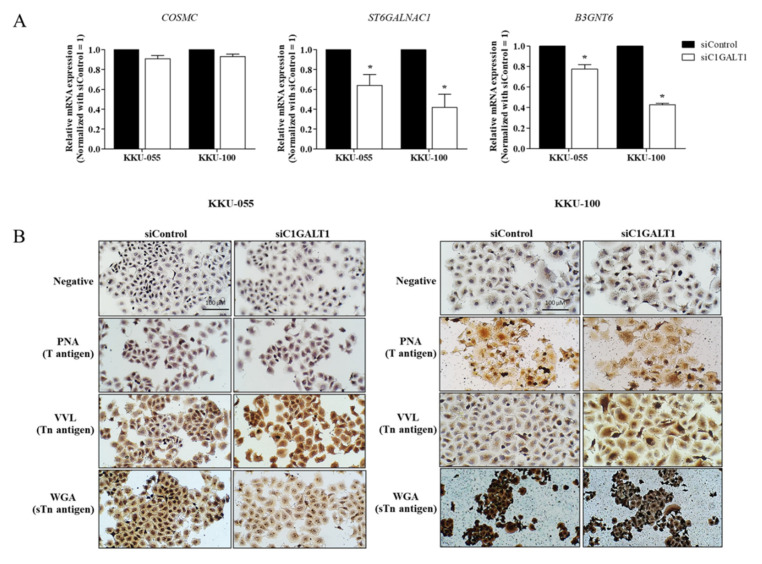
Silencing C1GALT1 is associated with immature mucin-type O-glycosylation. KKU-055 and KK-100 cells were treated with siC1GALT1 and siControl for 48 h. (**A**) The relative mRNA expression levels of *COSMC*, *ST6GALNAC1* and *B3GNT6* was performed using qPCR and compared between the C1GALT1 knockdown and control cells. Relative mRNA expression level was normalized to β-actin as the reference genes and calculated using the 2^−ΔΔCt^ method. Results were presented as mean ± SEM of three independent experiments. * *p* < 0.05 VS siControl. (**B**) Lectin-cytochemimical staining of PNA, VVL and WGA in C1GALT1 knockdown and control cells (magnification, ×200). Scale bar, 100 µm.

**Table 1 life-12-00174-t001:** Sequences of the primers used for reverse transcription-quantitative PCR.

Gene	Forward (5′-3′)	Reverse (5′-3′)
*C1GALT1*	5′-GGG AAT CTG GGC GGC A-3′	5′-GGG ACT GGT GAC CTT TGC TT-3′
*COSMC*	5′-AAC GTG AGA GGA AAC CCG TG-3′	5′-AAA GCA TTT TTC CCG CGT CT-3′
*B3GNT6*	5′-TCA ACC TCA CGC TCA AGC AC-3′	5′-CAG GAA GCG GAC TAC GTT GG-3′
*ST6GALNAC1*	5′-CAG AGG CAC AAT CAT GGA AG-3′	5′-GCT GAC TTT TGG GAA TGA GC-3′
*C* *-* *Myc*	5′-CTG CTG TGG ACC CTA CTG-3′	5′-AAC TGC GTC TCT GCC AGG AC-3′
*BIRC5*	5′-TGA GGA GAC ACC GCC CAC-3′	5′-CAA CAT CGA TTT CTT CCT CAT CTT C-3′
*ABCC1*	5′-CTG GGC TTA TTT CGG ATC AA-3′	5′-TGA ATG GGT CCA GGT TCA TT-3′
*ABCC3*	5′-CCT GCT CTC CTT CAT CAA TC-3′	5′-ATG TAG TGG TAA TAG TGT TGT AAG-3′
*ABCG2*	5′-CAC CTT ATT GGC CTC AGG AA-3′	5′-CCT GCT TGG AAG GCT CTA TG-3′
*CCND1*	5′-CCA CTT GAG CTT GTT CAC CA-3′	5′-AAC TAC CTG GAC CGC TTC CT-3′
*CDK4*	5′-GTC GGC TTC AGA GTT TCC AC -3′	5′-TGC AGT CCA CAT ATG CAA CA-3′
*CCNE1*	5′-GAA ATG GCC AAA ATC GAC AG -3′	5′-TCT TTG TCA GGT GTG GGG A -3′
*CCNB1*	5′-GAC AAC TTG AGG AAG AGC AAG C -3′	5′-ATG GTC TCC TGC AAC AAC CT -3′
*β* *-* *actin*	5′- GAT CAG CAA GCA GGA GTA TGA CG -3′	5′-AAG GGT GTA ACG CAA CTA AGT CAT AG-3′

## Data Availability

All data generated or analyzed during the present study are included in this published article.

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
