# Peer review of "Down-Regulation of C1GALT1 Enhances the Progression of Cholangiocarcinoma through Activation of AKT/ERK Signaling Pathways"

_life, 2022, doi:10.3390/life12020174_

Round 1
Reviewer 1 Report
In this manuscript, the authors aimed to explore the regulation of C1GALT1 in the progression of cholangiocarcinoma (CCA) in 2D cell cultures after observing the C1GALT1 downregulation in CCA tissues. Their results indicate that C1GALT1 silencing contributes to the formation of the immature truncated core 1 O-glycan, promoting CCA progression and 5-FU resistance by activating the AKT/ERK signaling pathway. The results may provide novel therapeutic strategy for CCA, for which an effective treatment is still necessary.
Overall, this study was well designed, well and clearly written and the findings are interesting.
However, the manuscript has some weaknesses, such as: the 2D model used in vitro instead of the 3D model more similar to the in vivo condition and the use of only two stabilized cell lines. Nevertheless, the data processing from patients and the use of CCA tissues for some experiments makes the manuscript closer to clinical reality.
Major points:
The authors need to clarify the following points and to strengthen the results with some experiment:
- What are the main characteristics of the cell lines used? Do they have characteristic genetic alterations?
- What are the clinical characteristics of patients who have high/low C1GALT1?
- 2D tumor progression study lacks cell migration/invasion tests
- What happens to the most aggressive cell lines if CGALT1 is over-expressed?
Minor points:
In the legend of figure 3 was reported "anti-apoptotic proteins", but there are no in the figure
Author Response
Dear: Life Editors and Reviewer,
Thank for your review of manuscript (Life-1552037) entitled, “Down-regulation of C1GALT1 enhances the progression of cholangiocarcinoma through activation of AKT/ERK signaling pathways”. Our responses to reviewer are on the attached file. Please see the attachment.

Reviewer 2 Report
In manuscript entitled „Down-regulation of C1GALT1 enhances the progression of 2 cholangiocarcinoma through activation of AKT/ERK signaling 3 pathways” the authors showed that down-regulation of C1GALT1 in CCA increases the expression of immature core 1 O-glycan, enhancing CCA progression, including growth and 5-fluorouracil resistance via the activation of the AKT/ERK signaling pathway.
The topic is very interesting, and the authors showed some interesting results. However, I have some suggestion for revision:
- Juthamas Khiaowichit1, Chutima Talabnin2*, Chavaboon Detchsukum3, Atit Silsirivanit4, Krajang Talabnin3* and – et the end of the authors list you put and. You should delete this.
- In the Material and Methods section the authors stated, “Informed written consent was obtained from all patients, and the Ethics Committee for Human Research from Khon Kaen University and Suranaree University of Technology reviewed and approved the procedure for sample collection (HE521209 and EC-57-25, respectively).” – while et the end of manuscript they stated different “Ethics approval and consent to participate: Not applicable.”
- In Material and Methods section – page 3, line 95 – Vector Laboratories.
- In Material and Methods section – page 4, line 127 – the authors stated “used using Hoechst 33258 DNA staining” – you should delete one word.
- In Material and Methods section – page 4, line 146 - the authors stated “Immunohistochemical staining was performed (28).” This should be clearly stated - Immunohistochemical staining was performed according to previous confirmed experimental procedure (28).
- In Material and Methods section – page 5, line 160 - the authors stated, “The 0 scores were categorized as C1GALT1-low expression, while scores of 1 and 2 were categorized as C1GALT1-high expression.” – This should be revised. You cannot claim that something that has a score of 0 has a low degree of expression. You should mark these samples as those that do not have expression, while those with 1 and 2 can be divided into two groups with low and high level of expression.
- All results presented in graphs you should present as a mean value (mean±SD) of your samples, not as percentage.
- Also, in Results page 10, line 261 the authors stated, “endogenous C1GALT1 expression was examined in four CCA lines (KKU-055, KKU-100, KKU-213A, and KKU-213B)” but in all other section you provide results only for two cell lines.
Author Response

(The authors gave the same response as above.)

Reviewer 3 Report
Treatment of advanced CCA is still limited. The authors describe the role of C1GALT1 in CCA and demonstrate in human samples and in preclinical in vitro experiments that downregulation of C1GALT1 contributes to a more aggressive phenotype and chemotherapy resistance. Overall, the study is well planned and conducted and results are clearly presented and discussed. The authors also highlight some limitations of their study, e.g. lack of functional in vivo data.
Some points should be adressed:
- Please be consistent if a cell viability or a cell proliferation assay was used in vitro.
- Please provide a kappa value for inter-observer variability on the assessment of IHC data.
- Why did you not use h score but only categorized data on expression in IHC?
- Fig 2C does not show a significant difference in OS. Is there data on PFS or disease free survival available? This could indicate on the putative role on aggressiveness of the target.
- Please provide densitometry analyses for Westernblot data.
- The 5-FU concentrations used in the in vitro experiments seem rather high (150 and 300 µM). Is there an intrinsic resistance to 5-FU known for the cell lines used? Could this resistance impact on the readouts related to the functional experiments
- Line 145: please correct µM (micro molar) to µm (micro meter)
- Figure 4A and C: b-actin loading controls for KKU-100 look identical. Was the analysis performed on the same blot/membrane?
Author Response

(The authors gave the same response as above.)

Round 2
Reviewer 1 Report
no comments